# Machine Learning Coronary Artery Disease Prediction Based on Imaging and Non-Imaging Data

**DOI:** 10.3390/diagnostics12061466

**Published:** 2022-06-14

**Authors:** Vassiliki I. Kigka, Eleni Georga, Vassilis Tsakanikas, Savvas Kyriakidis, Panagiota Tsompou, Panagiotis Siogkas, Lampros K. Michalis, Katerina K. Naka, Danilo Neglia, Silvia Rocchiccioli, Gualtiero Pelosi, Dimitrios I. Fotiadis, Antonis Sakellarios

**Affiliations:** 1Unit of Medical Technology and Intelligent Information Systems, Department of Materials Science and Engineering, University of Ioannina, GR 45110 Ioannina, Greece; kigkavaso@gmail.com (V.I.K.); egewrga@gmail.com (E.G.); vasilistsakanikas@gmail.com (V.T.); savvasik21@gmail.com (S.K.); panagiotatsompou@gmail.com (P.T.); psiogkas4454@gmail.com (P.S.); dimitris.fotiadis30@gmail.com (D.I.F.); 2Institute of Molecular Biology and Biotechnology, Department of Biomedical Research—FORTH, University Campus of Ioannina, GR 45110 Ioannina, Greece; 3Department of Cardiology, Medical School, University of Ioannina, GR 45110 Ioannina, Greece; lamprosmihalis@gmail.com (L.K.M.); drkknaka@gmail.com (K.K.N.); 4Fondazione Toscana Gabriele Monasterio, IT 56126 Pisa, Italy; dneglia@ftgm.it; 5Institute of Clinical Physiology, National Research Council, IT 56124 Pisa, Italy; silvia.rocchiccioli@ifc.cnr.it (S.R.); pelosi@ifc.cnr.it (G.P.)

**Keywords:** coronary artery disease, noninvasive cardiovascular imaging, coronary artery disease risk stratification, machine learning models

## Abstract

The prediction of obstructive atherosclerotic disease has significant clinical meaning for the decision making. In this study, a machine learning predictive model based on gradient boosting classifier is presented, aiming to identify the patients of high CAD risk and those of low CAD risk. The machine learning methodology includes five steps: the preprocessing of the input data, the class imbalance handling applying the Easy Ensemble algorithm, the recursive feature elimination technique implementation, the implementation of gradient boosting classifier, and finally the model evaluation, while the fine tuning of the presented model was implemented through a randomized search optimization of the model’s hyper-parameters over an internal 3-fold cross-validation. In total, 187 participants with suspicion of CAD previously underwent CTCA during EVINCI and ARTreat clinical studies and were prospectively included to undergo follow-up CTCA. The predictive model was trained using imaging data (geometrical and blood flow based) and non-imaging data. The overall predictive accuracy of the model was 0.81, using both imaging and non-imaging data. The innovative aspect of the proposed study is the combination of imaging-based data with the typical CAD risk factors to provide an integrated CAD risk-predictive model.

## 1. Introduction

Atherosclerosis is considered as a chronic inflammatory disease of arteries, and its clinical manifestation accounts for a significant number of deaths worldwide. Atherosclerotic disease is characterized by the pathologic process of lipid accumulation and inflammation in the vessel wall, leading to the vessel wall thickening, lumen stenosis, calcification, and in some cases thrombosis [1]. The most important form of atherosclerosis is coronary artery disease (CAD), which accounts for the largest portion of cardiovascular disease deaths and leads to narrowing of the arteries that carry blood to the heart muscle [2]. The recent advances in coronary imaging techniques, either invasive or noninvasive, have enabled the identification of coronary vessels features, which are considered as CAD risk factors. 

However, despite the recent technological cardiovascular imaging advancements to recognize the subclinical disease and the improvement of patient’s management, the identification of high-risk patients remains a challenge due to the inherently unpredictable disease’s nature [3] since the rate of major adverse cardiac events (MACE) remains high both for patients with known CAD or for asymptomatic individuals [4]. 

Both CAD risk prediction and its progression prediction are two issues of high importance in biomedical research that aims to identify those individuals who are associated with an increased risk of CAD and the main factors that contribute to the disease progression. Existing studies have reported the different types of CAD risk factors, such as the patient’s lipid profile (total cholesterol and low-density lipoprotein cholesterol, high-density lipoprotein cholesterol, triglycerides), smoking, hypertension, diabetes mellitus, obesity, and family history [5], and have established the importance of the conventional CAD risk factors in the prediction of CAD.

In the literature, different studies have been proposed for CAD risk prediction and the classification of patients into risk categories, either taking advantage of statistical modelling or artificial intelligence-based models [6,7,8]. The traditional statistical-based CAD prediction models have implemented regression models, such as the Cox model utilized in Framingham Risk Score study [6] and the Weibull model applied for the Systematic Coronary Risk Evaluation (SCORE) model. In spite of their predictability, statistical models are often dedicated to interpreting the input parameters and contributing to features association input analysis [9]. On the other hand, machine-learning-based models perform an automated search in the input features, either stochastic or deterministic, for the optimal prediction outcome and, in some cases, may be found advantageous over traditional regression methods [10,11].

As far as the existing machine-learning-based studies, different studies have been presented both for the prediction of CAD and the prediction of its progression, whereas other studies are dedicated to detect the most significant biomarkers. More specifically, Exarchos et al. [10] implemented typical classification schemes to predict the number of vessels’ stenosis, the atherosclerosis progression, as well as a hybrid score corresponding to the severity of the disease. The utilized input features were demographics, clinical data, several biochemical variables, monocytes, and adhesion molecules. In another recently published study [11], demographics, clinical data, echocardiography data, and 54 features of laboratory variables were used to predict the status of CAD by applying a support vector machine (SVM) algorithm with kernel fusion. Ambale et al. [12] implemented machine learning techniques to characterize cardiovascular risk, predict outcomes, and identify biomarkers in population studies. More specifically, they tested the ability of random forests (RF) to predict six different cardiovascular events and concluded that the RF technique performed better than established risk scores with increased prediction accuracy. Motwani et al. [13] found that machine learning techniques combining clinical and CTCA data predict 5-year all-cause mortality (ACM) in patients with suspected coronary artery disease better than existing clinical or CTCA metrics alone. In a study proposed by van Rosendael et al. [14], they investigated whether a machine-learning-based score incorporating only the 16-segment coronary tree information derived from CTCA provides enhanced risk stratification compared with current CTCA-based risk scores and concluded that the proposed model can improve the integration of CTCA-derived plaque information to improve risk stratification. In a more recent study, Sakellarios et al. [15] presented a multi-parametric predictive model, including traditional risk factors, plasma lipids, 3D imaging parameters, and computational data, for the prediction of site-specific plaque progression and concluded that imaging-based characteristics, such as low endothelial shear stress (ESS) and low-density lipid (LDL) accumulation, are significant predictors. On the other hand, Heo et al. [16] developed and validated machine learning models to predict patients with hidden CAD and assess long-term outcomes in patients with acute ischemic stroke.

The basic concept of the proposed study is to develop a machine learning predictive model that incorporates both noninvasive imaging data derived by CTCA and typical patient baseline characteristics to predict the CAD risk and especially the obstructive disease. The clinical focus of the proposed machine-learning-based model is to indicate the prognostic value of the combination of non-imaging and imaging features derived by CTCA imaging for the prediction of CAD high-risk patients and to compare the predictability of the combination of non-imaging and imaging features with non-imaging features alone. As for the innovative aspect of the proposed study, we assemble a variety of patient characteristics that have never been previously utilized, aiming to predict the risk for CAD. The final selected predictive model adopted both bagging and boosting ensemble modelling principles such that the model’s variance and bias are treated concurrently. 

## 2. Materials and Methods

### 2.1. Dataset Description

The proposed study is based on the EVINCI population [17], in which patient-specific information, both imaging and non-imaging, were collected for clinical purposes and utilized as the baseline information for the development of a CAD risk stratification methodology, whereas the follow-up data were collected after 6.22 ± 1.42 during the SMARTool project (September 2016–November 2017) [18]. More specifically, during the H2020 SMARTool project, a prospective, multicenter study in patients was conducted by 7 medical centers (Pisa, Turku, Zurich, Barcelona, Warsaw, Naples, Viareggio) from 5 European countries. All the participants signed informed consent to participate in the study and all the following procedures. Patients who previously underwent coronary CTCA during the EVINCI (Evaluation of Integrated Cardiac Imaging for the Detection and Characterization of Ischemic Heart Disease; FP7-222915; n = 152—February 2009–June 2012) [12] and ARTreat (FP7-224297; n = 18) [13] clinical studies were prospectively included to undergo follow-up CTCA. In addition to this, individuals (n = 32) who underwent CTCA in the period from 2009 to 2012 were also prospectively included. A detailed list of inclusion and exclusion criteria is provided in Appendix A.

Anonymized data were acquired from 187 patients, derived by different medical centers, and the cohort data were obtained under a data protection agreement fulling all the ethical and legal requirements for data sharing posed by the General Data Protection Regulation in a third-level care setting. Table 1 below demonstrates the collected data types. The median age of the patients of our dataset is 61 years old (45–76), and at their first visit to the physician, all the participants underwent CTCA imaging regardless of the presence of symptoms. More specifically, 45% and 25% of the participants had atypical and typical angina, respectively, whereas 12% of them had other symptoms, and 16% were asymptomatic. In addition to this, as for the pharmaceutical treatment of the participants, 18%, 28%, 13%, 40%, 13%, 10%, 3%, and 48% of them received angiotensin receptor blockers (ARBs), angiotensin converting enzyme inhibitors (ACE inhibitors), diuretics, beta blockers, calcium antagonists, oral antidiabetics, insulin, and statins, respectively, at the baseline time step.

### 2.2. Methodology

#### 2.2.1. CTCA Image Analysis and Three-Dimensional Reconstruction 

The first step of the development of the CAD risk-prediction model was the analysis of the CTCA images. This analysis was conducted by implementing an active contour based model for the segmentation of CTCA images and aimed to provide a detailed geometry of the three major coronary arteries, the left anterior descending artery (LAD), the left circumflex artery (LCX), and the right coronary artery (RCA). This methodology is integrated in a dedicated software tool, which can semi-automatically provide the detailed 3D coronary artery anatomy [19,20]. More details for the overall three-dimensional reconstruction methodology can be found in the Appendix A in the Section 2.2.1. 

#### 2.2.2. Calculation of the SmartFFR index

In this study, except the geometrical derived metrics and a blood-flow-based index, the SmartFFR index [21] was utilized. More details about the SmartFFR index can be found in the Appendix A, in the Section 2.2.2.

#### 2.2.3. Problem Definition

The CAD risk stratification problem has been formulated as a two-class classification problem based on the maximal coronary artery stenosis. This hypothesis is based on the findings of the Coronary Artery Disease Reporting and Data System (CAD-RADS) [22], which provides a standardized method to associate findings of the CTCA imaging modality to facilitate decision making regarding further patient management. Figure 1 shows the distribution of the population across the two CAD-severity groups. More specifically, among the total 263 patients who underwent CTCA imaging for clinical purposes, 55 patients underwent percutaneous coronary intervention stenting procedure and 10 patients coronary artery bypass grafting procedure, whereas CTCA images of 11 patients were considered as interpretable either at the baseline time step or at the follow-up time step. The annotation was based on the assessment of the obstructive disease: at least one major artery with stenosis > 50%. 

The definition of these two classes is based on the quantitative degree of stenosis derived by the CTCA imaging modality according to the society of cardiovascular computed tomography guidelines committee [23]. More specifically, the first class, the no CAD—minimal CAD class (Class 1$—C_{1}$)), includes the grading scale 0, 1, and 2 (normal, minimal, and mild), whereas the obstructive CAD class (Class 2$—C_{2}$)) includes the grading scale 3, 4, and 5 (moderate, severe, and occluded), as it is shown in Table 2. This classification was selected because we want to predict the obstructive CAD disease. 

Baseline imaging and non-imaging characteristics were trained into a gradient boosting classification scheme, aiming to discriminate the patients at low risk (Class $C_{1}$) and those at high risk (Class $C_{2}$), concerning their follow-up time step. This predictive supervised learning approach aims to learn mapping from input features x to output Y given a labeled set of input output pairs $D={\left\{\left(x_{i},Y_{i}\right)\right\}}_{i=1}^{N}$, where D is the training set, and N is the number of training examples [24]. Each sample $\left(x_{i},Y_{i}\right)$ associates the input features with the risk prediction of CAD severity, Y, where $Y\in \left\{C_{1},C_{2}\right\}$, is estimated by a non-linear parameterized function (f) of input features $x\in R^{d},$ $x=\left[x_{1},x_{2},\ldots ,\;x_{N}\right]$. The goal of this supervised classification problem is to obtain an approximation F(x) of the function $F^{*}\left(x\right)$ mapping the input x to output Y. The function $F^{*}\left(x\right)$ minimizes the expected value of some specified loss function L(y,F(x)), whereas the procedure followed in this proposed study is to restrict the function F(x) to be a member of parameterized class of functions F(x;Y). In addition to this, in this paper, we constructed our model based on additive expansions of the form $F\left(x;{\left\{\beta_{m},a_{m}\right\}}_{1}^{M}\right)$. $F^{*}\left(x\right)$ and $F\left(x;{\left\{\beta_{m},a_{m}\right\}}_{1}^{M}\right)$, which are described in the Appendix A, respectively [24].

The selected predictive model was nested into an easy ensemble classification scheme to overcome the class imbalance problem. To estimate the classification performance of the proposed method, an externally stratified 10-fold cross-validation was applied, with data pre-processing, a multivariate feature ranking, and a gradient boosting classification scheme being efficiently combined at each iteration of the procedure. The overall proposed model performs feature selection in the learning time since it achieves model fitting and feature selection simultaneously. Data-preprocessing and feature ranking follow the resampling procedure itself, which reduces the selection bias in the estimates of the model’s performance, whereas stratification assures that each validation fold retains the class distribution in the dataset. In addition to these, randomized search optimization of the model’s hyper-parameters over an internal 3-fold cross-validation contributes to the fine-tuning of the presented model.

#### 2.2.4. Easy Ensemble Algorithm Implementation-Class Imbalance Handling

The easy ensemble algorithm [25] is a class imbalance handling approach in which P are the training instances of the minority class, whereas Q denotes the instances of the majority class. The idea of the easy ensemble algorithm is to employ random resampling to generate *K* subsets of $\left\{Q_{1},Q_{2},\ldots .,Q_{K}\right\}$ from Q ($\left|Q_{i}\right|<\left|Q\right|$, *i* = 1, 2, …, *K*). Subsequently, each $Q_{i}\cup P$ is trained by the classifier, and the final decision is selected by majority voting. In the proposed predictive model approach, the easy ensemble approach is combined with the gradient boosting classifier, and each individual model is trained by the Appendix A.

#### 2.2.5. Data Pre-Processing

In this step, one hot encoding procedure was implemented to represent all the categorical input features as binary vectors. In addition to this, a curation procedure was implemented to curate our dataset both for outliers and missing values. All the input features whose missing values were higher than 10% were removed from the dataset, whereas features with missing values lower than 10% were imputed by either the most frequent value (categorical type features) or the median value (numerical type features).

#### 2.2.6. Recursive Feature Elimination

In this step, our aim is to reduce the dimensionality d of input features $x\in R^{d}$ to overcome the risk of overfitting, which basically arises when the number of d is comparatively large, and the number of the training patterns is small. In this study, a feature ranking technique with a support vector machine (SVM) with recursive feature elimination (RFE) was implemented to rank the input features. The whole SMV RFE procedure is shown in Appendix A [26].

#### 2.2.7. Gradient Boosting Classification

In the first step, the gradient boosting classification algorithm [27] implements a numerical implementation minimizing the equation of $F^{*}\left(x\right)$ (Appendix A). The whole function of the utilized classification scheme is described in the Appendix A.

The overall pipeline of the proposed machine learning methodology is shown in Figure 2 below.

## 3. Results

### CAD Risk-Prediction Model Performance Evaluation

The utilized CAD risk-prediction model performance metrics are the balanced accuracy, the negative predictive value, the positive predictive value, the area under the receiver operating curve (ROC AUC), and the sensitivity and specificity. The values of the adopted performance metrics and their mean value and the 10-fold standard deviation are given in Table 3. The average balanced accuracy of the selected predictive model is 0.81, while its sensitivity and specificity is 0.88 and 0.73, respectively. In Figure 3, we demonstrate the normalized confusion matrix regarding the selected gradient boosting classification algorithm combined with an SVM RFE feature selection technique. In addition to this, in Table 4, the respective performance metrics over the different folds and their mean and standard deviation values using only non-imaging data are shown. The average balanced accuracy of the predictive model trained only by non-imaging features is 0.69, while its sensitivity and specificity are both 0.69. 

Additionally, a SHAPley Additive exPlanations (SHAP) analysis was implemented for explaining the prediction of the proposed model by computing the contribution of each feature to the prediction [28]. The most important predictors of the proposed model are presented in Figure 4 below. Mean absolute SHAP values for the 10 most significant features are estimated to illustrate the global feature importance. As it is shown in Figure 4, the most significant feature is the number of the existing calcified plaques and the highest coronary degree of stenosis at the baseline step. In addition to this, input features such as pro-brain natriuretic peptide (NT-proBNP), matrix metalloproteinase-2 and 9 (MMP-2, MMP-9), leptin, low-density lipoprotein (LDL), and patient characteristics such as weight, age, and height are highly ranked as significant features for the prognosis of coronary artery disease (CAD).

In addition to this, in Figure 5 below, we demonstrate the global interpretability of the proposed model by representing how much each input feature, either positively or negatively, contributes to the target variable. In Figure 5, we show with yellow columns the input features that contribute positively to the output target (detection of Class 2, CAD class). On the other hand, with blue columns, we indicate the input predictors that contribute negatively to the output target (detection of Class 1, no-CAD class). As it is shown in Figure 5, most of the input features contribute negatively to the output target and contribute to the prognosis of Class 1. Indicatively, the most significant features that contribute positively to the output target are thyroid stimulating hormone, medication therapy of beta blockers, aspartate aminotransferase, diabetes, and minimum lumen area. In the presented model, we observe that the most significant predictor for the prognosis of CAD is thyroid stimulating hormone, which confirms the effect of the thyroid hormones on the cardiovascular system [29,30]. Thyroid hormone is considered is a significant regulator of cardiovascular system function and hemodynamics through different mechanisms. More specifically, inadequate thyroid hormone levels impair the relaxation of vascular smooth muscle cells and decrease cardiac contractility by regulating calcium uptake and the expression of several contractile proteins in cardiomyocytes. Additionally, low thyroid hormone levels also increase systemic vascular resistance and induce endothelial dysfunction by reducing nitric oxide availability [31,32]. As for the imaging-based input predictors, minimum lumen area has the most significant positive effect on the proposed model. As it is shown in Figure 5, the most significant feature with negative effect on the output is the number of the calcified plaques at the baseline analysis of patient imaging. Different studies in the literature have confirmed the prognostic capability of the presence of calcified atherosclerotic plaques [33]. Calcification of the coronary arteries plays a key role in the pathophysiology of atherosclerosis, and these lesions are considered advanced lesions [34]. In addition to this, patient height contributes negatively to the prognosis of the output target, confirming the genetic relationship between height and coronary artery disease [35,36]. As for the biochemical predictors for the no-CAD class (Class 1), we observed that pro-brain natriuretic peptide, low density lipoprotein, and matrix metalloproteinase 2 have a high negative effect on the output target.

## 4. Discussion

In this study, a novel approach for the prediction of obstructive CAD is presented. The aim of this study is to develop a machine learning model for the CAD risk prediction, which takes into account different types of data, including both imaging and non-imaging data. To our knowledge, our approach to combine the imaging and blood-flow-based characteristics with typical CAD risk factors constitutes the novelty of the presented study.

Different methodologies have been presented for the prediction of CAD and the identification of the major CAD risk factors. Most of these studies are concentrated on the different CAD-related risk-prediction outputs and are based either on statistical analysis [6,37,38] or machine learning classification schemes [39]. Our proposed study in comparison with these ones is more concentrated on the CAD risk prediction and its future presence and achieves a higher AUC.

Additionally, recent studies have indicated that non-invasive cardiovascular imaging and especially the CTCA imaging modality utilized in this study provides useful prognostic information of atherosclerosis progression since it permits the accurate quantification of luminal area and the detection of plaque burden region and the characterization of its composition. Moreover, the overall plaque burden, which can be provided by CTCA imaging, is highly relevant to the degree and characteristics of atherosclerosis [40]. In addition to this, the clinical relevance of the overall coronary plaque burden has been also emphasized by studies showing that increased non-calcified plaque volumes is directly linked with acute coronary syndrome (ACS) patients [41]. Furthermore, the latest technological advancements in patient-specific blood-flow modeling have introduced alternative CAD progression risk factors, such as fractional flow reserve (FFR) index and wall shear stress (WSS). 

The prognostic capability of CTCA imaging modality and its derived imaging features has also been confirmed by the proposed study, in which the overall accuracy of the proposed predictive model using both imaging and non-imaging data is 0.81. Moreover, the prognostic significance of imaging-derived features is also indicated by the collected results, shown in Table 4. More specifically, the predictive model trained by the non-imaging-based features achieved a comparatively lower accuracy of 0.69.

Furthermore, another notable point of the proposed CAD risk-predictive model is that the input geometrical features are derived by an automated CTCA image analysis tool [19,20], able to detect accurately the inner and outer wall and atherosclerotic plaques and provide an accurate 3D model of coronary arteries and the atherosclerotic plaques distribution over the 3D space. As far as the SmartFFR index is concerned, it is also calculated automatically by the developed software tool in the 3D-reconstructed coronary artery.

In addition to this, another innovative aspect of the presented predictive model is the implementation of the easy ensemble algorithm, which constitutes a random resampling scheme, which mainly handles the class imbalance problem. Except for the class imbalance handling, the applied easy ensemble scheme allows the progressive correction of the model’s decision hyperplane and subsequently the reduction of the classification error. In addition to this, the predictive capability of the proposed model is evaluated based on nested stratified cross-validation, which provides and unbiased estimation of the predictive model’s capability. Moreover, except for the innate hyperparameters of the classification algorithm, the input features are also treated as a hyper-parameter, and an SVM RFE feature selection technique is implemented to eliminate the input features’ dimension. The particular machine learning algorithm was selected after the implementation of different classification schemes in combination with different feature selection techniques, and the highest accuracy was provided by the combination of the extreme gradient boosting algorithm and the support vector machine (SVM) feature selection technique.

However, except for the prediction of CAD presence, the prediction of the CAD-related events is also a very important task both for the clinical research area and for patients’ management. However, the proposed methodology was trained using an existing dataset of 187 participants, in which there were only few CAD-related events. This is a low–medium-risk population, and we have few major CAD events to use for the development of such an event-prediction model. On the other hand, thanks to the advantage of our intermediate CAD risk population, we were able to build a model that can be used as a prognostic decision-support tool by clinicians to properly monitor and manage patients of intermediate CAD risk for the next years after a first imaging is available. 

A future step of the proposed methodology will include the integration of additional imaging-based features, which will be either based on CTCA image analysis or on blood-flow modeling. Additionally, another future step will be the development of predictive models aiming to predict CAD-related events either using imaging and non-imaging data and the investigation of the predictability of these features when the desired outcome is the CAD-related events. 

## 5. Conclusions

This proposed CAD risk-predictive model highlights the clinical utility of machine learning models to identify individuals of high CAD risk and those at risk of a potential CAD clinical event. In this investigation, we conclude that both imaging-derived features, combined with typical CAD risk factors and typical biochemical markers, have a significant predictive capability considering risk-prediction problem for CAD presence. In clinical practice, the utilization of such machine-learning-based approaches could improve CAD risk stratification and contribute to better strategies for patients’ management.

## Figures and Tables

**Figure 1 diagnostics-12-01466-f001:**
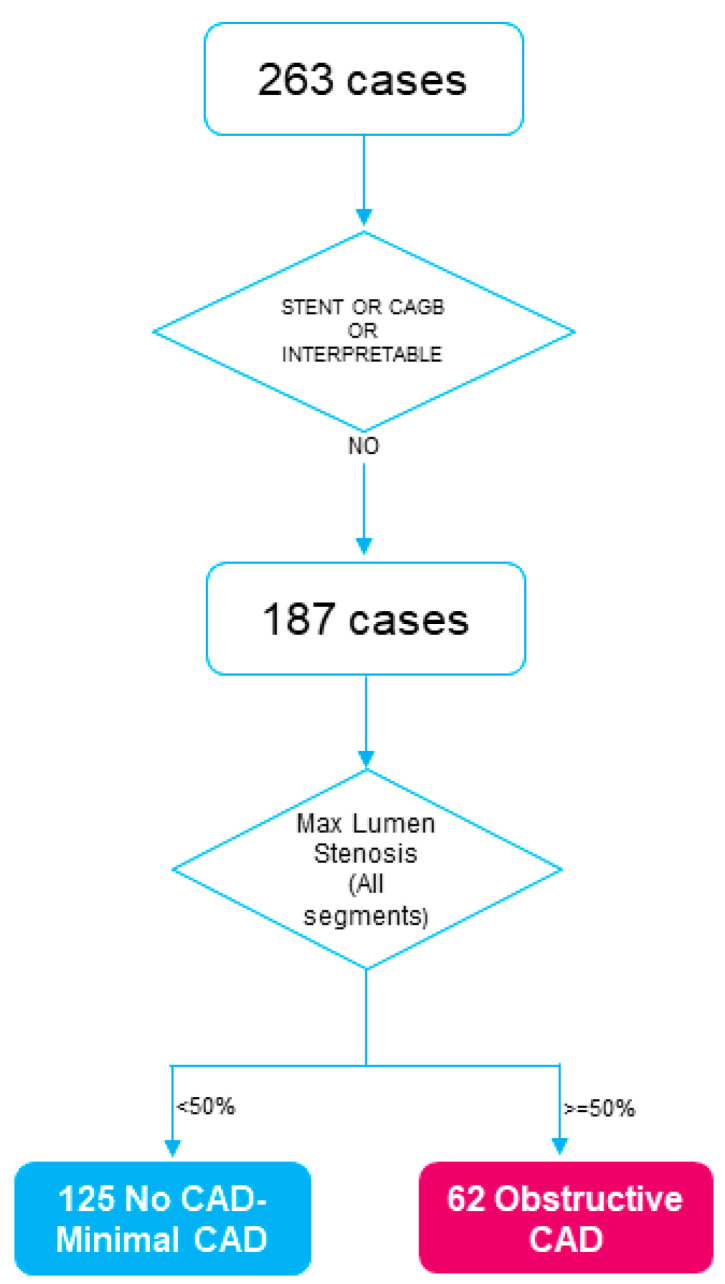
Flow chart depicting the distribution of the cohort in CAD-severity groups based on the CTCA imaging at the follow-up step. in total, 287 patient imaging data (125 in Class 1 and 62 in Class 2) were analyzed. (CAGB, coronary artery bypass graft surgery; CAD, coronary artery disease).

**Figure 2 diagnostics-12-01466-f002:**
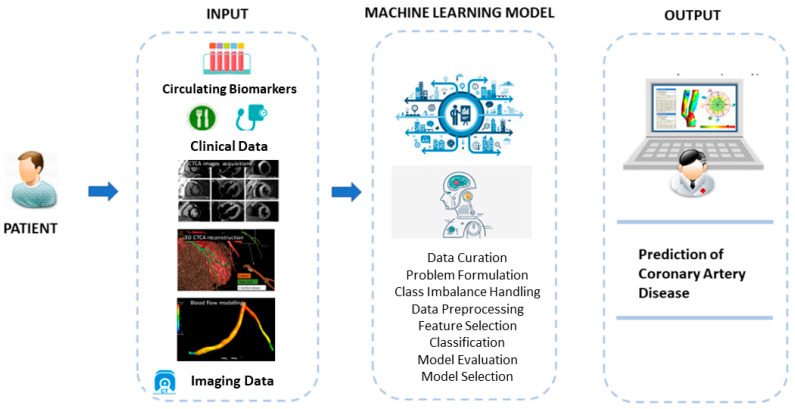
Overall pipeline of the proposed methodology. The input is based on clinical data, laboratory test, and imaging data provided by the three-dimensional reconstruction of the artery and the blood-flow modelling. Different machine learning models were implemented for the prediction of coronary artery disease presence.

**Figure 3 diagnostics-12-01466-f003:**
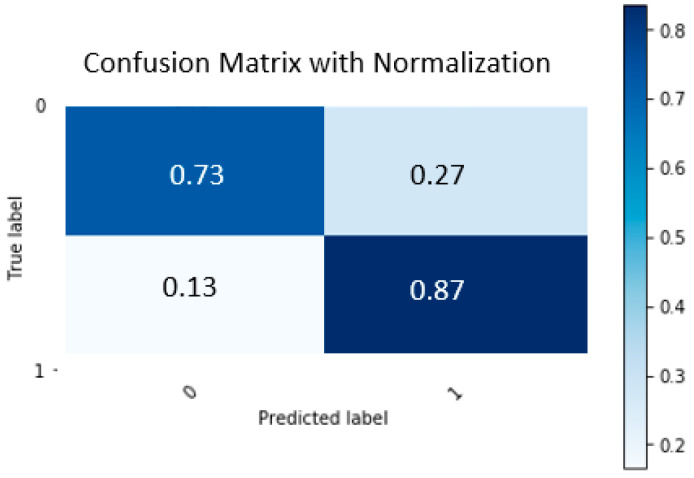
Normalized Confusion Matrix regarding the Gradient Boosting Classification algorithm for the CAD risk prediction using imaging and non-imaging data. The percentage of the true negative predicted cases is 73%, whereas the percentage of the true positives cases is 87%.

**Figure 4 diagnostics-12-01466-f004:**
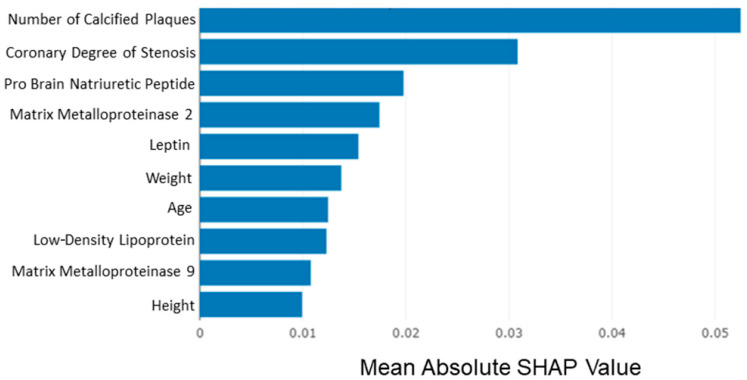
Feature importance based on mean SHAP values. The number of the existing calcified plaques and the highest coronary degree of stenosis are indicated as the most significant features.

**Figure 5 diagnostics-12-01466-f005:**
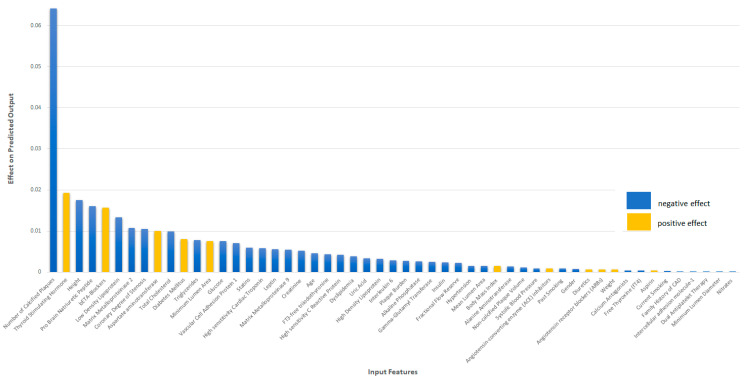
Input Features Contribution Table (blue, features with negative effect; yellow, features with positive effect). The most significant features that contribute positively to the output target are thyroid stimulating hormone, medication therapy of beta blockers, aspartate aminotransferase, diabetes, and minimum lumen area, whereas the most significant feature with negative effect on the output is the number of the calcified plaques at the baseline analysis of patient imaging.

**Table 1 diagnostics-12-01466-t001:** Imaging and non-imaging data utilized. * Imaging data from CTCA.

Type		Features
Imaging data *	Geometrical vasculature	Degree of Stenosis, Minimal Lumen Area, Minimal Lumen Diameter, Plaque Burden, Calcified Plaque Volume, Noncalcified Plaque Volume, SmartFFR Index, Number of Calcified Plaques, Number of Non-calcified Plaques
Non-imaging data	Demographics	Age, Gender
	Risk factors	Family History of CAD, Hypertension, Diabetes, Dyslipidemia, Smoking, Obesity, Metabolic Syndrome, Past Smokers
	Biohumoral Markers	Creatinine, Uric Acid, Glucose, Total Cholesterol, HDL, LDL, Triglycerides, Insulin, Aspartate Aminotransferase, Alanine Aminotransferase, Alkaline Phosphatase, Gamma-glutamyl Transferase, Hs-C Reactive Protein, Interleukin-6, TSH, fT3, fT4, Leptin, MMP2 Protein Plasma, MMP9 Protein Plasma, hs-cardiac Troponin T, N terminal Fragment of Pro-brain Natriuretic Peptide, Lipidomics, Metabolomics

**Table 2 diagnostics-12-01466-t002:** Definition of the utilized CAD risk classes. (CAD, coronary artery disease).

Proposed Classes	Recommended Stenosis Grading Scale of CAD	Quantitative Stenosis
$\mathrm{Class}\;1—C_{1}$	0: Normal	No luminal stenosis
	1: Minimal	Plaque with <25% stenosis
	2: Mild	25–49% stenosis
$\mathrm{Class}\;2—C_{2}$	3: Moderate	50–69% stenosis
	4: Severe	70–99% stenosis
	5: Occluded	100% stenosis

**Table 3 diagnostics-12-01466-t003:** Evaluation of the CAD risk-prediction problem over 10-fold using imaging and non-imaging data (AUC, area under curve).

Folds	Balanced Accuracy	Negative Predictive Value	Positive Predictive Value	ROC AUC	Sensitivity	Specificity
Fold #0	0.73	0.78	0.67	0.60	0.67	0.78
Fold #1	0.75	0.86	0.63	0.82	0.84	0.67
Fold #2	0.89	1	0.72	0.92	1	0.78
Fold #3	0.69	0.78	0.6	0.72	0.6	0.78
Fold #4	0.84	1	0.63	0.83	1	0.67
Fold #5	0.84	1	0.63	0.89	1	0.67
Fold #6	0.95	1	0.84	1	1	0.89
Fold #7	0.78	1	0.56	0.78	1	0.56
Fold #8	0.8	0.86	0.72	0.88	0.84	0.75
Fold #9	0.8	0.86	0.72	0.75	0.84	0.75
Mean ± std	**0.81 ± 0.08**	**0.92 ± 0.1**	**0.68 ± 0.08**	**0.82 ± 0.11**	**0.88 ± 0.15**	**0.73 ± 0.09**

**Table 4 diagnostics-12-01466-t004:** Evaluation of the CAD risk-prediction problem over 10-fold using only non-imaging data (AUC, area under curve).

Folds	Balanced Accuracy	Negative Predictive Value	Positive Predictive Value	ROC AUC	Sensitivity	Specificity
Fold #0	0.5	0.6	0.4	0.33	0.33	0.67
Fold #1	0.56	0.64	0.5	0.65	0.33	0.78
Fold #2	0.72	1	0.5	0.89	1	0.44
Fold #3	0.69	0.78	0.6	0.69	0.6	0.78
Fold #4	0.79	0.88	0.67	0.87	0.8	0.78
Fold #5	0.78	1	0.56	0.78	1	0.56
Fold #6	0.79	0.88	0.67	0.8	0.8	0.78
Fold #7	0.72	1	0.5	0.76	1	0.44
Fold #8	0.6	0.64	0.67	0.79	0.33	0.88
Fold #9	0.71	0.75	0.67	0.83	0.67	0.75
Mean ± std	**0.69 ± 0.1**	**0.82 ± 0.16**	**0.57 ± 0.1**	**0.74 ± 0.16**	**0.69 ± 0.28**	**0.69 ± 0.15**

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
