# Peer review of "Machine Learning Coronary Artery Disease Prediction Based on Imaging and Non-Imaging Data"

_diagnostics, 2022, doi:10.3390/diagnostics12061466_

Round 1

Reviewer 1 Report

I thank the authors for having solved the raised issues. I do not have any further comments. 

Please note that a warning about missing reference is still present.

Author Response

We thank the reviewer for the comment.  We have revised the manuscript’s references according to the references section of the manuscript submission guidelines.

Reviewer 2 Report

Dear Author,

The authors have suggested a machine learning model for the CAD risk prediction utilizing CTCA and non-imaging data including demographics and risk factors and biomarkers. 

  1. The author did not provide a figure legend and sufficient explanation of each result, or full name of abbreviations.
  2. How to distinguish between Symptomatic and Asymptomatic in 62 obstructive patients?
  3. Can you provide how to analyze non-calcified plaque volume and the number of non-calcified plaques on CTCA images?
  4. Extensive editing is needed such as submission format.

Author Response

The authors have suggested a machine learning model for the CAD risk prediction utilizing CTCA and non-imaging data including demographics and risk factors and biomarkers.

Comment #1: The author did not provide a figure legend and sufficient explanation of each result, or full name of abbreviations.

We have revised the manuscript according to reviewer’s comment.  More specifically, all the legends of the figures have been modified by adding additional explanation including the full name of all the abbreviations.

Comment #2: How to distinguish between Symptomatic and Asymptomatic in 62 obstructive patients?

During the H2020 SMARTool project, a prospective, multicenter study, baseline information of all the participants were collected.  The data of the patients were both imaging and non-imaging data.  As for the non-imaging data, clinical data based on the presence of symptoms at the baseline step were collected .  45% and 25% of the participants have atypical and typical angina, respectively, whereas 12% of them had other symptoms and 16% were asymptomatic, regardless the future presence of coronary artery disease.  The aim of this study is not to predict either the symptomatic or asymptomatic patients, but to identify the participants with high risk of coronary artery disease presence based on the degree of stenosis.  Additionally, the number of the participants with obstructive disease is limited in order to distinguish between symptomatic and asymptomatic patients.

Comment #3: Can you provide how to analyze non-calcified plaque volume and the number of non-calcified plaques on CTCA images?

The entire methodology for the CTCA images analysis has been presented in detailed in two different publications, cited in the manuscript and reported in the Appendix [1, 2].

As for the non-calcified plaques (NCP) segmentation, the process has been achieved by a dynamic thresholding technique. The threshold value is not considered as a constant and relies on the mean intensities values of pixels, which correspond to the lumen region.  The main idea of this approach is to define the intensity range of the dynamic threshold technique around the luminal intensity and the range of NCP HU values is extracted based on the mean luminal intensity (ml). The ml is computed after the implementation of the Frangi vesselness filter. The ml value is the mean value of the highest of half image intensities, which are higher than 100 HU, considering only the parts of the CTCA image, which are potential coronary vessels. After the definition of the ml, the range of NCP pixels intensities is defined from 100 HU to the ml value. In addition to this, the aforementioned segmentation approach is implemented in the ROI, which is located between the segmented outer wall and lumen.  After the NCP segmentation, a surface construction process follows, based on the Marching cube algorithm [1, 2].

Comment #4: Extensive editing is needed such as submission format.

We thank the reviewer for this comment. Editing of the manuscript was implemented following the submission guidelines.

Round 2

Reviewer 2 Report

The authors have presented a machine learning model for the CAD risk prediction using imaging and non-imaging data which an accuracy is 0.81. Comments are below

1. In the method, the authors described 'The median age of the patients of our dataset is 61 years old (35-76)' but aged 45-82 were included In the inclusion criteria in an appendix. It is necessary to clarify these sentences over 76 were included or not. Furthermore, It is necessary to clarify between  negative effect on the output such as number of calcified plaques and positive effect on the output such as TSH.  In addition, would be better to add sentences about how TSH and calcification are associated with the prediction. 

2. Input:  initialize:  Repeat until S=( )...

are empty after sentences in table 1, appendix 

3. References formatting is different in an appendix and the main context and required to establish formatting such as m3 converted to m3

4. The authors concluded that the CAD risk-related machine learning models can identify the individuals with high CAD risk and of those of a potential CAD clinical event. However, the authors did not describe CAD events association related to prediction factors. 

3. Why CACS over 600 has excluded from the data?

Author Response

The authors have presented a machine learning model for the CAD risk prediction using imaging and non-imaging data which an accuracy is 0.81. Comments are below:

Comment #1: In the method, the authors described 'The median age of the patients of our dataset is 61 years old (35-76)' but aged 45-82 were included in the inclusion criteria in an appendix. It is necessary to clarify these sentences over 76 were included or not.

Actually, the median age of the participants in the utilized dataset is 61 years old and the age of the patients ranges from 45 to 76 years old.  The number 35 was a typographical error. We have revised our manuscript in line 129:

The median age of the patients of our dataset is 61 years old (45-76) and at their first visit to the physician, all the participants underwent CTCA imaging, regardless the presence of symptoms.

Comment #2: Furthermore, it is necessary to clarify between negative effect on the output such as number of calcified plaques and positive effect on the output such as TSH.  In addition, would be better to add sentences about how TSH and calcification are associated with the prediction.

According to this comment, we have modified the manuscript and we have added the following lines:

lines 293-299: Thyroid hormone is considered is a significant regulator of cardiovascular system function and hemodynamics through different mechanisms.  More specifically, inadequate thyroid hormone levels impair the relaxation of vascular smooth muscle cells and de-crease cardiac contractility by regulating calcium uptake and the expression of several contractile proteins in cardiomyocytes.  Additionally, low thyroid hormone levels also in-crease systemic vascular resistance and induce endothelial dysfunction by reducing nitric oxide availability.

lines 303-306: Different studies in the literature have confirmed the prognostic capability of the presence of calcified atherosclerotic. Calcification of the coronary arteries plays a key role in the pathophysiology of atherosclerosis and these lesions are considered advanced lesions.

Comment #3: Input: initialize: Repeat until S=( )... are empty after sentences in table 1, appendix

In Table 1 in the Appendix, we show the SVM Recursive Feature Elimination (SVM RFE) approach which is utilized, as it is reported in Guyon et al. [1] study.

Comment #4: References formatting is different in an appendix and the main context and required to establish formatting such as m3 converted to m3.

We thank the reviewer for this comment.  We have modified the references’ style of the Appendix, based on the writing guidelines and we have modified the format of m3 to m3.

Comment 5:  The authors concluded that the CAD risk-related machine learning models can identify the individuals with high CAD risk and of those of a potential CAD clinical event. However, the authors did not describe CAD events association related to prediction factors.

As for the CAD related clinical events, in the utilized dataset we study the future CAD presence and not the future cardiac events.  More specifically, in lines 375-390, we report the following for this comment:

However, except of the prediction of CAD presence, the prediction of the CAD related events is also a very important task both for the clinical research area and for the patients’ management.  However, the proposed methodology has been trained using an existing dataset of 187 participants in which there were only few CAD related events.  This is a low-medium risk population and we have few major CAD events to use for the development of such an event prediction model. On the other hand, thanks to the advantage of our population that is of intermediate CAD risk, we are able to build a model which can be used as a prognostic decision support tool by the clinicians to properly monitor and manage patients of intermediate CAD risk for the next years after a first imaging is available.

Our future step of the proposed methodology will include the integration of additional imaging based features, which will be either based on CTCA image analysis or on the blood flow modeling.  Additionally, another future step will be the development of predictive models aiming to predict CAD related events either using imaging and non-imaging data and the investigation of the predictability of these features when the desired outcome is the CAD related events. 

Comment #6: Why CACS over 600 has excluded from the data?

This exclusion criterion has been defined by the clinical doctors, which have conducted the clinical study, since very high values of CAC score will be make difficult the CTCA image processing.

References

  1. Guyon, I.; Weston, J.; Barnhill, S.; Vapnik, V. Gene selection for cancer classification using support vector machines. Machine learning 2002, 46, 389-422.

This manuscript is a resubmission of an earlier submission. The following is a list of the peer review reports and author responses from that submission.

Round 1

Reviewer 1 Report

In this paper the authors describe a Machine Learning CAD prediction based on Imaging and non-Imaging data. The topic is of sure interest, both for the risks related to CAD progression and for the novelty of the methodology. However, some issues need to be solved in order to make this paper sound and useful for the reader.

  1. In my opinion, introduction is too long. Some concepts related to CAD are out of topic and repeated. Furthermore, lines 132-147 should be moved in methods or results.
  2. Conversely, results section is too short. Considering the wide description of the methodology, I would have expected more details and explanations, in order to better clarify the importance of the obtained results. This section need to be expanded.
  3. Authors compare the performance of their model with other ones available in literature. However, information reported in table 5 are related to the original performance of those models, obtained in a different setting, thus not comparable with the setting of the model proposed by the authors. I personally would remove this table as misleading.

Minor issues:

Please check warning about reference source not found across the text.

Reviewer 2 Report

Dear authors:

  • Review references ( a lot of mistakes)
  • Rewritte Matherial and methods (it is too hard to read, make it simply)
  • No data about events